# JOINT DENOISING OF CRYO-EM PROJECTION IMAGES USING POLAR TRANSFORMERS

## ABSTRACT

Deep neural networks (DNNs) have proven powerful for denoising individual images, but there is a limit to the noise level they can handle. In applications like cryogenic electron microscopy (cryo-EM), the noise level is extremely high but datasets contain hundreds of thousands of projections of the same molecule, each taken a different viewing direction. This redundancy of information is useful in traditional denoising techniques known as class averaging methods, where images are clustered, aligned, and then averaged to reduce the noise level. We present a neural network architecture based on polar representation of images and transformers that simultaneously clusters, aligns, and denoises cryo-EM projection images. Results on synthetic data show accurate denoising performance using this architecture, with a relative mean squared error of $0.06$ at signal-to-noise (SNR) level of $0.05$, outperforming traditional filter-based methods by a factor of $2\times$.

## 1 INTRODUCTION

Many imaging problems amount to reconstructing some unknown quantity (often two-dimensional images or three-dimensional density maps) from several measurements obtained through a lossy, noisy forward process. In the simplest case where the forward process is identical for all measurements, information from these measurements can be combined by simple averaging, which reduces the relative noise level. This suggests using similar strategies also for the general case where there is some difference in setup for each measurement.

Cryo-EM (Dubochet et al., 1982) is a molecular imaging technique that uses transmission electron microscopes to analyze the structure of molecules, typically proteins of interest for biological or medical research. In its single-particle reconstruction (SPR) form, this involves freezing a sample containing millions of copies of a single molecule in a thin layer of vitreous ice, imaging the frozen sample in an electron microscope, and then reconstructing the 3D structure of the molecule from the resulting 2D projection images, known as *micrographs*. This process involves first extracting images of individual particles from the micrographs, denoising these images, estimating their corresponding viewing directions, and then reconstructing a 3D density map (Frank, 1996).

Apart from the unknown viewing directions, a main challenge of cryo-EM imaging is the high noise level. Indeed, since the electron dose needs to be kept low enough to avoid radiation damage during the imaging process, each projection contains only a weak signal besides a large amount of shot noise (Bendory et al., 2020). Denoising is therefore a central task in any cryo-EM pipeline, such as for visual inspection (Singer & Sigworth, 2020), detecting contamination (outliers) (Bhamre et al., 2016), generating templates for particle picking (Singer & Sigworth, 2020), identifying high-quality projection images (Scheres, 2015), and as preprocessing for certain ab-initio reconstruction methods (Singer & Shkolnisky, 2011). However, despite the large number of molecules in the sample, averaging is not directly applicable because of the unknown viewing directions.

Traditional denoising methods can be divided into two categories. The first is based on applying a Wiener filter to the images (Frank et al., 1996; Tang et al., 2007; Sindelar & Grigorieff, 2011; Bhamre et al., 2016). While this approach exploits prior information on the content of the images and is optimal in the class of linear estimators, it often results in a high level of blurring for the relevant noise levels. The second group of denoising methods is known collectively as *class averaging* methods. These rely on the geometry of the problem described above – given the large number of images, some will be taken from the same viewing direction and only differ by an in-plane rotation.

Class averaging methods therefore first classify the images, where each cluster consists of images that are similar up to some in-plane rotation, then align those images and average them (van Heel, 1984; Park & Chirikjian, 2014; Zhao & Singer, 2014; Scheres, 2015). This average has less noise, no bias introduced by a filter, and can then be used for subsequent reconstruction tasks. However, the high noise level often makes both classification and alignment very challenging (Jensen, 2001).

Like many other disciplines, cryo-EM processing has been improved in several ways by the introduction of methods from deep learning. This ranges from particle picking (Wang et al., 2016; Wagner et al., 2019) and postprocessing of 3D densities (Sanchez-Garcia et al., 2021) to using implicit neural representations to represent 2D images (Bibas et al., 2021; Nasiri & Bepler, 2022; Kwon et al., 2023) and 3D structures (Zhong et al., 2021; Nashed et al., 2021; Levy et al., 2022; Schwab et al., 2024). DNNs have also been used for denoising, but in the setting of entire micrographs (Bepler et al., 2020; Buchholz et al., 2019; Palovcak et al., 2020). This problem is different from the denoising of projection images in several respects, such only being able to leverage information within a single micrograph. These methods also do not explicitly incorporate rotational symmetries of the data distribution, limiting their performance.

This work proposes to solve the denoising problem by simultaneously denoising multiple images in a manner that extends traditional class averaging by leveraging prior information on the content of the projection images. The novel architecture rotationally aligns and integrates information from multiple cryo-EM projection images using a new convolutional attention mechanism that also guarantees the overall rotational equivariance of the system. This network, known as *polar transformer*, is shown to successfully cluster, align, and denoise sets of cryo-EM images on simulated datasets, yielding a reduction of more than $2\times$ compared to classical Wiener filter methods on images of SNR as low as $0.05$, or $1.5\times$ compared to deep neural networks operating on single projections.

The rest of this paper is structured as follows. Section 2 describes the three main denoising tasks: single-image denoising, aligned denoising, and class averaged denoising. Then Section 3 reviews some of the existing literature on the subject of cryo-EM denoising. Sections 4 and 5 describe the construction of the polar transformer and some of its properties. Finally, section 6 gives details of concrete models and and how we trained and tested them.

## 2 PROBLEM SETUP

The denoising problem in cryo-EM aims at recovering a clean image $x \in X$ from a noisy version $y \in X$. We represent all images through their pixel values on an $L \times L$ raster, i.e., $X = \mathbb{R}^{L \times L}$.

The clean images are assumed to be tomographic projections of a 3D potential density from a biological macromolecule. These are calculated using line integrals of that density along some chosen viewing direction. In the full model, there is also the application of a contrast transfer function (CTF) to simulate the optical properties of the microscope, but we will omit this in the present work to simplify the discussion. We shall also assume that these are images of individual projection images (i.e., we are past the particle-picking stage) and that images have been centered, but not rotationally aligned. While simple, the above model has proven to be relatively accurate (Frank, 1996; Vulović et al., 2013) and is used in a across range of cryo-EM reconstruction methods (Barnett et al., 2017; Scheres, 2012; Punjani et al., 2017; Zhong et al., 2021). To simulate the noisy projections $y$ from the clean image $x$, we then use additive Gaussian noise, a common assumption in many reconstruction methods (Punjani et al., 2017).

We will now consider three distinct denoising tasks.

### 2.1 INDIVIDUAL PROJECTIONS

The simplest setting is to consider each projection separately, potentially combining with information from other projections later on. We thus have a mapping $f : X \to X$ such that $x \approx f(y)$. We call this the *individual image denoising task*. This is the setting of Wiener filters, which are trained on a large set of images to calculate their statistics and then applied separately to each noisy image.

One property that we would like $f$ to exhibit is equivariance to rotations. This follows from the fact that the probability distribution of the images (both clean and noisy) is invariant to in-plane rotation. Indeed, one image is as likely to appear in the data as a rotated copy of that image. Letting

$R_\alpha : X \to X$ denote in-plane rotation of an image by $\alpha \in [0, 2\pi)$, we thus require

$$f(R_\alpha x) = R_\alpha(f(x)). \tag{1}$$

## 2.2 DIRECTIONAL SETS

More potent processing is possible when using a *set* of $K$ images, $\boldsymbol{x} \in X^K$, representing a selection of projections picked from a microscopy sample. In particular, we shall assume that these images all come from the same viewing direction and that they only differ by in-plane rotation (in other words, there is $\alpha_{ij}$ such that $x_i = R_{\alpha_{ij}} x_j$ for each $i, j$). We call this the *directional set denoising task*. This situation arises, for example, during class averaging (see below) when the clustered images are to be aligned and averaged.

Here, we require a similar equivariance property, but extend it so that each image in the set is rotated by a different angle. In other words, we have the mapping $R_{\boldsymbol{\alpha}} : X^K \to X^K$ such that $(R_{\boldsymbol{\alpha}} \boldsymbol{x})_i = R_{\alpha_i} x_i$. The equivariance relation in this case is

$$f(R_{\boldsymbol{\alpha}} \boldsymbol{x}) = R_{\boldsymbol{\alpha}}(f(\boldsymbol{x})). \tag{2}$$

## 2.3 GENERAL SETS

In the general case, we cannot assume that our images have already been clustered by viewing direction. Instead, we rely on the fact that any sufficiently large set already contains such clusters of images that are the same up to rotation. This is the underlying idea of class averaging (see below) and represents the most typical denoising setting. We will refer to this as the *full set denoising task* and the challenge here is identify the similar images in the set (up to rotation) and denoise them jointly. Such a denoiser again takes the form $f : X^K \to X^K$ and satisfies the same equivariance property equation 2 as in the directional set case.

## 3 RELATED WORK

Existing approaches to image denoising can be categorized into shallow (non-neural) approaches and DNNs. While the former often rely on geometrical properties of the images (Dabov et al., 2007), they also exploit (often implicitly) some prior knowledge about those images. DNNs have typically extended these methods by enriching the latter component, providing stronger priors encoded in the neural networks by training on a large set of images. In contrast, these networks are often less able to exploit geometrical properties of the images, such as those found in cryo-EM.

The simplest way to denoise cryo-EM images is to filter them. These can either be fixed low-pass filters (referred to as "binning") (Bartesaghi et al., 2018), stationary Wiener filters (Frank et al., 1996; Tang et al., 2007; Sindelar & Grigorieff, 2011) or generalized Wiener filters calculated in a steerable basis Bhamre et al. (2016). While these have achieved a certain degree of success, they are fundamentally limited in that they are *linear* denoisers, and can therefore only achieve optimality with respect to a Gaussian prior distribution. Since the prior distribution of clean cryo-EM images is necessarily non-Gaussian, these methods will typically result in clean, but overly blurred images.

Another method for denoising in cryo-EM is class averaging (van Heel, 1984; Park & Chirikjian, 2014; Zhao & Singer, 2014; Scheres, 2015). The central idea here is to use the fact that all the images in a cryo-EM dataset cluster naturally by viewing direction. Indeed, given a particular projection image, it is typically possible to find another set of images taken from the same viewing direction, but subject to a different in-plane rotation. The goal of these class averaging methods is therefore to cluster the images, align them, and then average. This results in images that have less blurring compared to filter-based methods, but often a large number of images (on the order of 100) are needed to achieve a low reconstruction error. Furthermore, the classification and alignment steps are quite sensitive to noise and therefore fail at low SNR.

Another way to improve over the linear denoisers is to consider non-linear denoisers. This is the basis of the DnCNN architecture (Zhang et al., 2017), where a standard residual CNN is trained to recreate the clean image from a noisy version of that image. While the work focused on denoising of photos, the technique can be applied to any set of images.

(a)       (b)           (c)           (d)

Figure 1: (a) A simulated $64 \times 64$ projection image of PDB ID 2pkq. (b) The projection image with a polar grid superimposed (downsampled by a factor of four for visualization purposes). (c) The polar representation. The horizontal axis represents angles and the vertical axis represents radii. (d) The reconstructed image with relative mean squared error $3.5 \cdot 10^{-4}$.

A related architecture are U-Nets (Ronneberger et al., 2015; Gurrola-Ramos et al., 2021). In the context of cryo-EM images, they have been applied for denoising, but only in the context of entire micrographs (Bepler et al., 2020; Palovcak et al., 2020). Though this has the advantage over our approach of not requiring particle picking beforehand, it puts the burden of exploiting any redundancy between the particles entirely on the trainable network. At low SNR and low information content of an individual projection image, training this is challenging, whereas our method hard-bakes it efficiently into the architecture (section 5).

More generally, DNNs have been used in the cryo-EM reconstruction pipeline to provide stronger priors on the 3D reconstruction. For example, Kimanius et al. (2021) replaced the weak stationary Gaussian process used in the RELION (Scheres, 2012) software with a learned prior. This resulted in an increase in accuracy, but also showed signs of hallucination. More recently, this approach has been extended by Kimanius et al. (2024), where the method was used to reconstruct very small molecules (of molecular weight 40 kDa).

In a separate line of research, DNNs have found widespread use in cryo-EM as a general-purpose function approximation tool. Here, DNNs are used in an unsupervised manner and optimized to fit a particular cryo-EM dataset, for example with the goal of fitting a 3D model (Zhong et al., 2021; Gupta et al., 2021; Schwab et al., 2024; Nashed et al., 2021; Levy et al., 2022) or the set of 2D projections (Bibas et al., 2021; Kwon et al., 2023; Nasiri & Bepler, 2022). Consequently, the neural networks typically do not impose a specific prior on the reconstructions beyond the inductive bias provided by the particular architecture. This is in contrast with the approach of Bepler et al. (2020); Kimanius et al. (2021; 2024) and the method proposed in this work, which train DNNs to form universal estimators by encoding prior information extracted from publicly available datasets.

## 4 POLAR DECOMPOSITION AND CNNS

For the task of denoising images, a natural approach is to train a convolutional neural network for the purpose, with noisy images as input and clean images as targets (Zhang et al., 2017; Gurrola-Ramos et al., 2021). While these have been successful in various domains, such as natural and medical images, they do not apply directly to cryo-EM projection images.

The main reason for this is that a single cryo-EM image has a much higher level of noise compared to other modalities. We thus need to denoise multiple images *jointly* by classifying, aligning, and averaging them in the manner of class averaging (van Heel, 1984; Park & Chirikjian, 2014; Zhao & Singer, 2014; Scheres, 2015) To achieve this using a neural network architecture, we need a representation that allows us to rotate images in an efficient and stable manner. As it turns out, this representation will also enforce rotational equivariance, which is desirable in a cryo-EM denoiser since the distribution of images is invariant to rotation.

### 4.1 POLAR REPRESENTATION

To achieve the properties discussed above, we propose to decompose the images using a weighed polar representation. While the original images are given in the Cartesian domain, with pixel values on an $L \times L$ grid, we will map those images to a polar grid consisting formed as the tensor product of a radial and an angular grid.

First, let us assume that the pixel size of the images corresponds to $2/L$, which means that the $L \times L$ Cartesian grid spans the square $[-1, 1]^2$ (see Figure 1a). We now want to inscribe a polar grid inside this square. The radial points $r_0, r_2, \ldots, r_{N-1}$ are given by a Gauss–Jacobi quadrature rule over $[0, \sqrt{2} + \Delta]$ with $\alpha = 0$, $\beta = 1$ and $N = L$ points (Ralston & Rabinowitz, 2001, Chapter 4.8-1) for some $\Delta \geq 0$. Let us denote the corresponding quadrature weights by $w_1, w_2, \ldots, w_N$. The angular points are given by $\alpha_m = 2\pi m/M$ for $m = 0, 1, \ldots, M = 4L$. The resulting grid $\{(u_{nm}, v_{nm})\}_{nm}$ is then given by the points (see Figure 1b)

$$u_{nm} = r_n \cos \alpha_m \quad v_{nm} = r_n \sin \alpha_m. \tag{3}$$

To map our Cartesian images with pixel values $x[i, j]$ to the polar domain, we place a Gaussian radial basis function (RBF) at each of the grid points and use this to weight the pixel values. Let us assume that $L$ is even and that the Cartesian image $x$ is indexed by $-L/2, \ldots, L/2 - 1$ along both axes. The polar decomposition is then given by (see Figure 1c)

$$Px[n, m] = Z^{-1} \sqrt{w_n} \sum_{i,j=-L/2}^{L/2-1} x[i, j] \exp\left(-\frac{(u_{nm} - 2i/L)^2 + (v_{nm} - 2j/L)^2}{2b^2}\right) \tag{4}$$

for $n = 0, 1, \ldots, N - 1$ and $m = 0, 1, \ldots, M - 1$, where $b$ is a bandwidth factor typically set to $1/L$ and the normalization factor $Z$ is given by

$$Z = \sqrt{2\pi^3 M L^2 b^4}. \tag{5}$$

This representation enjoys a number of useful properties. To see this, we apply the adjoint of equation 4 to some polar image $z[n, m]$, obtaining

$$P^{\mathrm{T}} z[i, j] = Z^{-1} \sum_{n=0}^{N-1} \sum_{m=0}^{M-1} \sqrt{w_n} z[n, m] \exp\left(-\frac{(u_{nm} - 2i/L)^2 + (v_{nm} - 2j/L)^2}{2b^2}\right) \tag{6}$$

This does not recover the original image $x$, but the following proposition shows that it can be approximated as a convolution for $N$ and $M$ large enough (for proof, see Appendix A).

**Proposition 1.** *Let*

$$\phi[i, j] = \frac{1}{\pi b^2 L^2} \exp\left(-\frac{i^2 + j^2}{b^2 L^2}\right). \tag{7}$$

*Then*

$$P^T Px = x \star \phi + O\left(\frac{(\sqrt{2} + \Delta)^2}{N^2 b^4} + \left(\frac{M!}{2M!}\right)^2 \left(\frac{2}{b^2}\right)^M + b e^{-\Delta^2/b^2}\right), \tag{8}$$

*where $\star$ denotes a discrete 2D convolution.*

Consequently, we can reconstruct $x$ by solving the deconvolution problem in equation 8. This can be done in several ways, most easily by approximating the discrete convolution with a circular convolution and solving it by pointwise division in the Fourier domain (Bertero et al., 2021). To ensure that this deconvolution problem is relatively well-posed, however, the Fourier transform of $\phi$ must not decay too fast – in other words, we cannot choose $b$ to be too small. We have found that choosing $b$ on the order of $1/L$ results in a well-conditioned deconvolution problem. The polar representation thus allows for an accurate, efficient, and stable reconstruction (see Figure 1d).

Second, above result shows that $P$ is an approximate isometry when restricted to smooth images. Indeed, we can show that $\phi$ is a lowpass filter which sums to one (for details, see Appendix B). As a result, we have that for $x$ smooth, then $x \star \phi \approx x$. This in turn then means that

$$(Py)^{\mathrm{T}}(Px) = y^{\mathrm{T}} P^{\mathrm{T}} Px \approx y^{\mathrm{T}}(x \star \phi) \approx y^{\mathrm{T}} x. \tag{9}$$

for two images $x$ and $y$. Because of this approximate isometry property, the polar mapping will not significantly distort the geometry of the original data manifold.

Another useful property of the above construction is its stability. Indeed, since the Fourier transform of $\phi$ has its largest value close to one, the largest eigenvalue of $P^{\mathrm{T}}P$ is also close to one, which means that $\|P\| \approx 1$. In other words, small changes in $x$ result in small changes in $Px$. This is

in contrast with other polar representations, such as those obtained by nearest-neighbor or linear interpolation, which can introduce artifacts for small changes in the Cartesian image. This lack of artifacts simplifies training of neural networks in the representation, improves their equivariance properties, and increases their robustness to noise.

In terms of computational cost, application of $P$ and $P^\mathrm{T}$ can be implemented efficiently using fast Gaussian gridding (Greengard & Lee, 2004; Barnett et al., 2019; Shih et al., 2021). As a result of this, the number of non-negligible terms in equation 4 and equation 6 is $\mathcal{O}(1)$, which means that both operations can be computed in $\mathcal{O}(L^2)$ time (recall that $N = \mathcal{O}(L)$ and $N = \mathcal{O}(L)$). Finally, the deconvolution step, if implemented using FFTs, has a computational cost of $\mathcal{O}(L^2 \log L)$. Consequently, mapping between Cartesian and polar domains can be achieved quite efficiently.

## 4.2 POLAR CNNS

Another important property of the polar representation is that it commutes with rotation. Let $R_\alpha$ denote rotation of an image by the angle $\alpha$ using some suitable interpolation scheme. Then we have the relationship $PR_{\alpha_\ell} \approx S_\ell P$, where $S_\ell$ denotes circular shift along the second axis by $\ell$ and we recall that $\alpha_\ell = 2\pi\ell/M$. In other words, for some image $x$, we have

$$PR_{\alpha_\ell}x[n,m] \approx S_\ell Px[n,m] = Px[n,m-\ell], \tag{10}$$

where the angular index is taken modulo $M$. Rotation in the Cartesian domain thus corresponds to translation along the angular axis in the polar domain. Due to the stability of the polar mapping $P$, off-grid rotations will give sensible polar representations in the form of sub-sample shifts.

If we want to construct a neural network that is equivariant to rotations in the Cartesian domain, this translates to equivariance to circular translation in the polar domain. The space of linear operators that are equivariant to circular translations along the angular axis is spanned by *angular convolutions*, that is, operators of the form

$$z \circledast h[n,m] = \sum_{p=0}^{N-1} \sum_{q=-Q}^{Q} z[p,m-q]h[n,p,q], \tag{11}$$

where, again, angular indices in $Px$ are taken modulo $M$ and $Q$ is the angular width of $h$ (typically in the range 1–3). We refer to $h$ in the above convolution as an *angular filter*, which has two radial indices $n, p$ and one angular index $q$.

Note that the operation above only convolves along the angular axis. Along the radial axis, it amounts to a matrix multiplication (i.e., a fully connected layer). Since we have no natural group action along this axis (we could consider scaling, but this would require a different grid configuration), we cannot simplify the operator further. However, we can enforce locality of the filter by restricting its support. Specifically, we set $h[n,p,q] = 0$ whenever $|n-p| > W$, for some radial width $W$ (typically in the range 1–3).

The angular filter can be trivially generalized to multiple channels and be used as a linear layer in a DNN. Since the polar representation is essentially a remapping of the original Cartesian image, standard layers from CNNs can be applied to the output of the angular convolutions, such as ReLUs (Goodfellow et al., 2016), avoiding less natural spectral non-linearities required for other equivariant networks (Kondor et al., 2018). On the other hand, the above construction also avoids costly transformations between an equivariant basis and a natural image basis (Cohen et al., 2018). Similar issues arise when attempting to construct DNNs based on steerable Fourier–Bessel bases (Zhao & Singer, 2013; Langfield et al., 2022). Finally, we note that the above construction can be formulated as an SO(2)-CNN in the formalism of Kondor & Trivedi (2018).

We shall refer to a DNN whose linear layers consist of the angular convolutions described above as *polar CNNs*. As constructed, they are equivariant to shifts along the angular axis and may be converted to a rotationally equivariant DNN operating on Cartesian images using the Cartesian-to-polar and polar-to-Cartesian mappings discussed above. We shall abuse terminology slightly and also refer to these as polar CNNs.

## 5 POLAR TRANSFORMER

To go beyond single-image denoising, we must find a way to combine information from multiple images. This is particularly relevant in the cryo-EM setting, where copies of the same molecule are imaged from different directions. Indeed, this observation is the basis for one of the most common denoising methods in cryo-EM, class averaging (van Heel, 1984; Park & Chirikjian, 2014; Zhao & Singer, 2014; Scheres, 2015).

One direct extension of the single-image denoiser would be to use its output in a class averaging method, where the denoised images are clustered and aligned. However, since training would not be end-to-end, the results would be suboptimal. In particular, the single-image denoiser is optimized to reproduce the original image, potentially discarding information relevant for clustering and alignment. We will instead combine denoising and class averaging into an end-to-end neural network that learns how to optimally cluster and align the images.

### 5.1 ATTENTION MECHANISM

A natural architecture for aggregating disparate sources of information is the transformer, originally introduced in the field of language models (Vaswani et al., 2017; Devlin et al., 2018; Brown et al., 2020), but which has since seen applications in image processing (Dosovitskiy et al., 2021), molecular modeling (Jumper et al., 2021), and other areas (Peebles & Xie, 2023; Ma et al., 2024). Here, a sequence of tokens are used to generate a key, query, and value vector that are combined using what is known as an *attention mechanism* (Vaswani et al., 2017). In language models, tokens are parts of words, but these can be any information carrier, such as an image.

Let us consider a multi-channel polar image $z[c, n, m]$, where $c = 0, 1, \ldots, C - 1$ is the channel index, $n = 0, 1, \ldots, N - 1$ is the radial index, and $m = 0, 1, \ldots, M - 1$ is the angular index. Now we consider three polar CNNs, denoted $f_\theta^{\text{key}}$, $f_\theta^{\text{query}}$, and $f_\theta^{\text{value}}$, corresponding to the key, query, and value networks, respectively. Here, $\theta$ is a vector of weights shared between the different networks.

Now suppose we have a set $z$ of $K$ such images $z_0, z_1, \ldots, z_{K-1}$, which we process using the above networks to key, query, and value vectors. Combining the key and query images, we obtain the scaled dot-product *attention coefficient*

$$\alpha_{k,k'}(z) = \text{softmax}\left(\frac{1}{\sqrt{CNM}} \sum_{c,n,m} f_\theta^{\text{query}}(z_k)[c, n, m] f_\theta^{\text{key}}(z_{k'})[c, n, m]\right) \quad (12)$$

for all $k, k = 0, 1, \ldots, K - 1$, where the softmax is taken with respect to second ($k'$) axis. These are then used to generate the $k$th output using the value vectors, yielding

$$f_\theta^{\text{attention}}(z) = \sum_{k'=0}^{K-1} \alpha_{k,k'}(z) f^{\text{value}}(z_{k'}). \quad (13)$$

If $f_\theta^{\text{attention}}(z)$ is trained using sequences of noisy and clean images with an MSE loss, it can learn to combine features from the various images in order to reduce the estimation error.

### 5.2 ANGULAR ATTENTION

While the standard attention mechanism may be useful for combining information in the images by clustering them appropriately, it has two important drawbacks: it requires the images to be rotationally aligned and it is not equivariant to rotations. In fact, it turns out that these problems are closely related – an attention mechanism that is rotationally equivariant must also perform alignment.

To see how this can be achieved, let us consider an augmented form of the attention mechanism, where each key is rotated by an arbitrary angle $\alpha_\ell$, that is, shifted by $\ell$ along the angular axis. We then have a set of rotated attention coefficients

$$\alpha_{k,k'}^{(\ell)}(z) = \text{softmax}\left(\frac{1}{\sqrt{CNM}} \sum_{c,n,m} f_\theta^{\text{query}}(z_k)[c, n, m] f_\theta^{\text{key}}(z_{k'})[c, n, m - \ell]\right), \quad (14)$$

for $\ell = 0, 1, \ldots, M - 1$, where the softmax is now taken over all indices $k'$ and $\ell$. We then apply a corresponding rotation to the value vectors when performing the linear combination, to obtain

$$f_\theta^{\text{ang-attention}}(z) = \sum_{\ell=0}^{M-1} \sum_{k'=0}^{K-1} \alpha_{k,k'}^{(\ell)}(z) S_\ell f_\theta^{\text{value}}(z_{k'}) \tag{15}$$

for $k = 0, 1, \ldots, K - 1$.

While this would seem to come at a significant computational expense, both calculation of the augmented attention coefficients and the output images can be written as convolutions along the angular axis. As a result, both can be implemented efficiently using FFTs.

Since it performs attention not only across the images in the input set, but along the angular axis, we refer to this mechanism as *angular attention*. It gives the network the ability to align images rotationally and combine them accordingly.

We can also show that it satisfies an extended equivariance property. Let $S_\ell$ denote the joint angular shifting operator for $K$ images by $\boldsymbol{\ell} \in \mathbb{Z}^K$, which shifts the $k$th image by $\ell_k$ along the angular axis. It can then be shown that

$$f_\theta^{\text{ang-attention}}(S_{\boldsymbol{\ell}} z) = S_{\boldsymbol{\ell}} f_\theta^{\text{ang-attention}}(z). \tag{16}$$

In other words, we can shift each polar image independently of one another and the resulting output will be shifted by the same amounts. In the Cartesian domain, this means that we have joint equivariance to rotation of the individual images in the set, i.e., the network is $\text{SO}(2)^K$-equivariant. This property is desirable since it means our denoiser is compatible with the invariance properties of the joint distribution of projection images (see equation 2).

## 6 EXPERIMENTS AND RESULTS

To evaluate the architectures proposed above, we conduct numerical experiments on simulated data. These show how the polar CNN and polar transformer are able to achieve very low denoising errors despite high noise levels, significantly outperforming baseline methods.

### 6.1 NEURAL NETWORKS

Two neural network architectures are used in the experiments. The first is a simple polar CNN consisting of a Cartesian-to-polar layer, 25 angular convolutional layers with 8 channels each, each followed by a GroupNorm layer (with 4 groups) (Wu & He, 2018) and a ReLU nonlinearity (Goodfellow et al., 2016). Finally, the polar representation was converted back to the Cartesian domain. The convolutional layers had an angular kernel width of 5 and a radial kernel width of 3.

The other network used was a polar transformer, which first consisted of a Cartesian-to-polar layer followed by a polar CNN preprocessing network of depth 5 applied individually to each image in the sequence. This was followed by an angular attention module where the key and query networks consisted of the same polar CNN of depth 3 with shared weights (the value network was set to the identity). Finally, the output of the attention module was postprocessed in a 9-layer polar CNN and the result converted back to the Cartesian domain. The convolutional layers in the CNNs had the same configuration as in the first architecture, but the number of channels in the pre- and postprocessing networks was set to 8 while that number was 16 for the key- and query-network.

### 6.2 DATA

Both architectures were trained on simulated data obtained from 5 000 molecular structures downloaded from the PDB (wwPDB consortium, 2018). Each molecule is projected through 1 000 different viewing directions to yield a total of 5 000 000 clean projection images. The same process was repeated for another set of 100 molecules, resulting in a testing set of 500 000 clean images. Both of these were then used to create three different datasets for training and testing, in accordance with the three denoising tasks described in Section 2.

For individual projection denoising, we simply added noise to each image at a specific SNR. For directional set denoising, each clean image was rotated through a random angle, resulting in $K = 8$

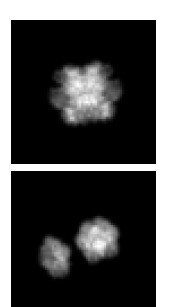 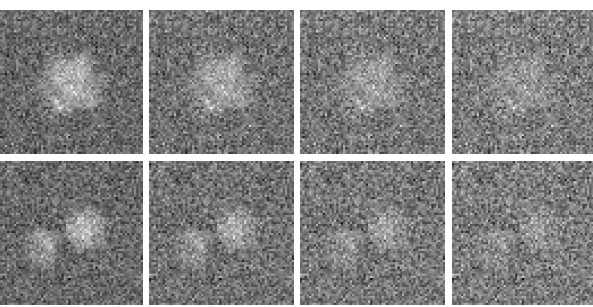 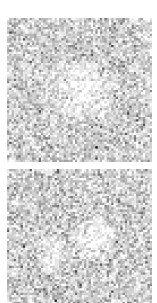

Figure 2: Two example projections in different noisy realizations of the same projection. Left: clean. Middle: with additive Gaussian noise at SNRs $\frac{1}{4}$, $\frac{1}{8}$, $\frac{1}{16}$ and $\frac{1}{32}$. Right: with Poisson noise, SNR $\frac{1}{32}$.

different clean images, which were then subjected to noise. This set of $K$ images was then used as input to the transformer model. In the general set denoising task, two clean images were picked corresponding to two different viewing directions, each of which was rotated to yield $K = 8$ copies. Each image was then subjected to noise. Both sets of copies were then concatenated to form a set of $2K$ images that was used as input to the transformer model.

To add the noise, we considered two approaches: Gaussian and Poisson. For Gaussian noise, we used white Gaussian noise at a fixed noise level that was added to the images. In addition to the noise level, the Poisson noise also included a *Poissonicity* parameter $\eta$ which controlled the extent to which it approached a Gaussian noise (which occurred as $\eta \to 0$) (for details, see Appendix C). Some sample images are shown in Figure 2.

## 6.3 TRAINING AND TESTING PROCEDURES

Each model was trained using the training sets described above for a specific SNR (defined as the average square magnitude of the clean images divided by the noise variance). While the models can be trained for a range of SNRs and produce adequate results, for simplicity we present results for fixed-SNR models in the current work.

All architectures were trained to minimize mean squared error loss using the Adam optimizer with a learning rate of $10^{-3}$ (Kingma & Ba, 2017) and a batch size of $128$. Convergence was typically obtained after ten epochs for all models.

The Wiener filter was constructed by estimating the mean and covariance over the entire training set in accordance with Bhamre et al. (2016). It was then applied to each image in the testing dataset with the corresponding noise level used to calibrate the filter. The DnCNN model follows Zhang et al. (2017). The U-Net corresponds to one also used in the Topaz pipeline (Bepler et al., 2020). Group normalization had to be added to avoid explosion of weights at low SNR. We trained these models in the same manner as our polar CNN.

## 6.4 RESULTS

The results in Figure 3 show that the the polar CNN network outperforms the Wiener filter, especially at high SNRs, where the error is reduced by a factor of two. At low SNR, most of the fine-scale structure in the projection images is destroyed by the noise, and therefore the CNN performance converges to that of the linear model. A similar behavior is observed for the other single-image denoisers, the DnCNN and U-Net, which do not explicitly encode rotational equivariance (and therefore perform slightly worse) but learn it through the data augmentation implicit in the training data.

Going beyond the polar CNN, we see that the polar transformer models consistently outperform single-image methods. For an SNR of $1/64$, we obtain a factor of two reduction compared to the Wiener filter in the directional set task.

Finally, we see that the polar transformer is also able to cluster sets of noisy images in the general set setting. While performance is strictly worse, it remains quite close to the directional set results,

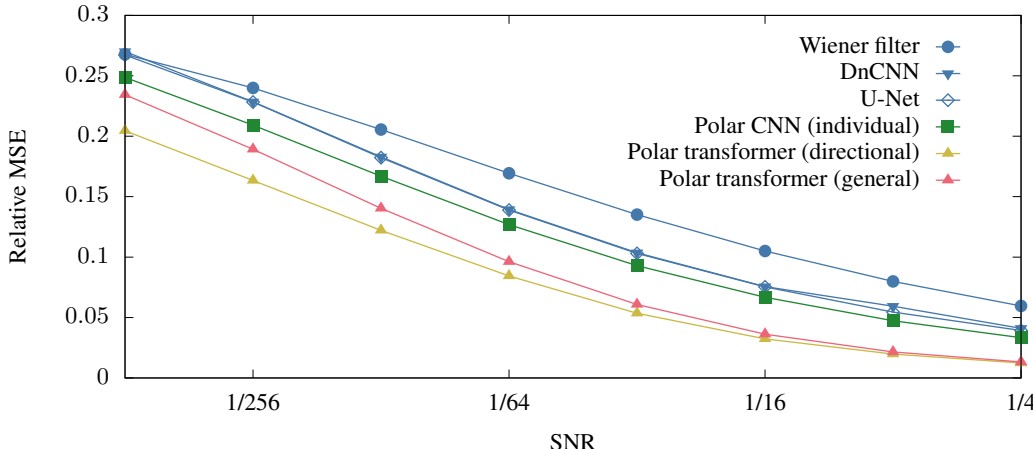

Figure 3: The denoising MSE for SNRs of Gaussian noise. The Wiener filter and polar CNN denoise individual images, while the transformers denoise sets of images (the directional set contains 8 images while the general set contains 16 images split evenly over two viewing directions).

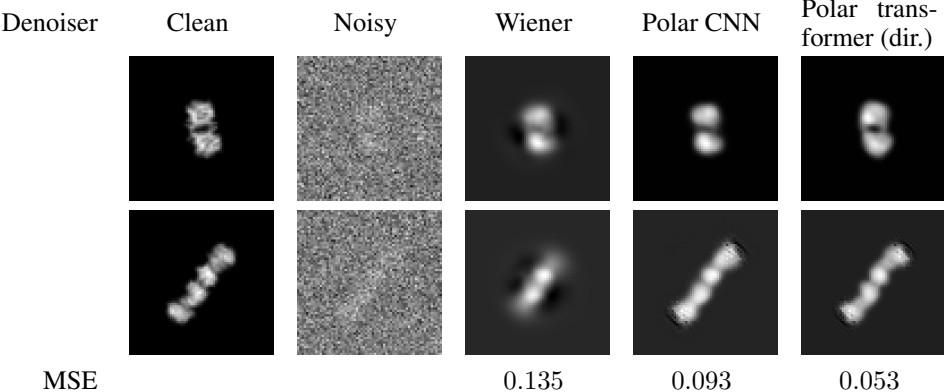

| Denoiser | Clean | Noisy | Wiener | Polar CNN | Polar transformer (dir.) |
|---|---|---|---|---|---|
| MSE | | | 0.135 | 0.093 | 0.053 |

Table 1: The performance of the denoisers for an SNR of $1/32$. Projections with additive Gaussian noise, processed by the polar transformer model on a directional set and two individual-projection denoisers ones for comparison. See table 2 for the Poisson noise case.

with a 15% drop in MSE at SNR $= 1/64$ and still well ahead of the baseline methods. Looking at the activation coefficients of the attention module (see Appendix D), we see that these indeed cluster the different viewing directions as expected.

We also see in the example images of Table 1 how the polar CNN produces sharper denoised images compared to the Wiener filter, and that the polar transformer in turn produces even less blurring. Finally, we see that the models are not very sensitive to the noise distribution in that models trained on Gaussian noise can also be applied successfully to images subjected to Poisson noise (see Appendix C. There is a small loss in MSE, but the overall features are preserved.

# 7 CONCLUSION

In this work, we have presented a new powerful architecture for joint denoising of cryo-EM projection images: the polar transformer. While this model has significant potential, more work remains before it can be applied to practical problems. Notably, they must be extended to incorporate point spread functions, translations, more realistic noise processes, unsupervised training (see discussion in Appendix E), and most importantly, scale to larger datasets.

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

# A   PROOF OF PROPOSTION 1

To see why $P^{\mathrm{T}}P$ can be approximated with a discrete convolution, let us write out the full expression

$$
P^{\mathrm{T}}Px[p,q] = Z^{-2} \sum_{n=0}^{N-1} w_n \sum_{m=0}^{M-1} e^{-\frac{(u_{nm}-2p/L)^2+(v_{nm}-2q/L)^2}{2b^2}} \sum_{i,j=-L/2}^{L/2-1} x[i,j] e^{-\frac{(u_{nm}-2i/L)^2+(v_{nm}-2j/L)^2}{2b^2}}
$$

$$
= Z^{-2} \sum_{i,j=-L/2}^{L/2-1} x[i,j] \sum_{n=0}^{N-1} w_n \sum_{m=0}^{M-1} e^{-\frac{(u_{nm}-2p/L)^2+(v_{nm}-2q/L)^2+(u_{nm}-2i/L)^2+(v_{nm}-2j/L)^2}{2b^2}},
$$

$$\tag{17}$$

where $p,q = -L/2, \ldots, L/2 - 1$.

If we denote the sum over $n, m$ by $M_{pqij}$ and use the identity

$$
(a-c)^2 + (b-c)^2 = \frac{1}{2}(a-b)^2 + 2\left(\frac{a+b}{2}-c\right)^2, \tag{18}
$$

we obtain

$$
M_{pqij} = \sum_{n=0}^{N-1} w_n \sum_{m=0}^{M-1} e^{-\frac{(2i/L-2p/L)^2+(2j/L-2q/L)^2}{4b^2} - \frac{((i+p)/L-u_{nm})^2+((j+q)/L-v_{nm})^2}{b^2}} \tag{19}
$$

$$
= 2\pi M e^{-\frac{(2i/L-2p/L)^2+(2j/L-2q/L)^2}{4b^2}} \sum_{n=0}^{N-1} \sum_{m=0}^{M-1} \frac{w_n}{2\pi M} e^{-\frac{((i+p)/L-u_{nm})^2+((j+q)/L-v_{nm})^2}{b^2}}. \tag{20}
$$

We now set $x = (i+p)/L$ and $y = (j+q)/L$ and note that $(x,y) \in [-1,1]^2$. Recall that we have $u_{nm} = r_n \cos(\alpha_m)$ and $v_{nm} = r_n \sin(\alpha_m)$ as described in equation 3. The sum over $n$ and $m$ above can therefore be written as

$$
\sum_{n=0}^{N-1} \sum_{m=0}^{M-1} \frac{w_n}{2\pi M} e^{-\frac{(x-r_n \cos(\alpha_m))^2+(y-r_n \sin(\alpha_m))^2}{b^2}}. \tag{21}
$$

We first consider the sum over the angular index $m$ for a fixed $n$. This is a sum of the periodic function $s_n(\alpha) = e^{-\frac{(x-r_n \cos(\alpha))^2+(y-r_n \sin(\alpha))^2}{b^2}}$ sampled on a uniform grid over $[0, 2\pi)$ of size $M$. Using a Fourier series decomposition of $s_n(\alpha)$, we can see that this sum is equal to

$$
\int_0^{2\pi} s_n(\alpha) d\alpha + \epsilon, \tag{22}
$$

where $|\epsilon| \leq \frac{C}{N^2} \int_0^{2\pi} |s_n''(\alpha)| d\alpha$ for some constant $C$. Differentiating $s_n(\alpha)$ and using the fact that $r_n < \sqrt{2} + \Delta$, we have that $|s_n''(\alpha)| \leq C(\sqrt{2} + \Delta)r_n$ for some constant $C$. This then gives

$$
\sum_{m=0}^{M-1} \frac{1}{2\pi M} e^{-\frac{(x-r_n \cos(\alpha_m))^2+(y-r_n \sin(\alpha_m))^2}{b^2}} = \sum_{m=0}^{M-1} \frac{1}{2\pi M} s(\alpha_m) \tag{23}
$$

$$
= \int_0^{2\pi} s_n(\alpha) d\alpha + \epsilon(r_n) \tag{24}
$$

where $|\epsilon(r)| \leq C(\sqrt{2} + \Delta)r/b^4 N^2$.

We now multiply by $w_n$ and sum over $n$. Since the error term $\epsilon(r)$ can be bounded by a linear function in $r$ and the Gauss–Jacobi quadrature integrates polynomials of degree $2M - 1$ exactly, we have

$$
\left| \sum_{m=0}^{M-1} w_n \epsilon(r_n) \right| \leq \frac{C(\sqrt{2} + \Delta)}{b^4 N^2} \int_0^{\sqrt{2}+\Delta} r^2 dr = \frac{C(\sqrt{2} + \Delta)^4}{b^4 N^2}. \tag{25}
$$

What remains is thus to calculate

$$\sum_{n=0}^{N-1} w_n I(r_n), \tag{26}$$

where $I(r) = \int_0^{2\pi} s_n(\alpha) d\alpha$. For the Gauss–Jacobi quadrature rule used here, the error is proportional to

$$\frac{M!}{2M!} \left( \frac{(M+1)!}{(2M+1)!} \right)^2 I^{(2M)}(r) \tag{27}$$

for some $r \in [0, \sqrt{2} + \Delta]$. Our goal is therefore to bound the $2M$th derivative of $I(r)$.

We first note that $(x, y)$ can be written in a polar representation, obtaining $x = \rho \cos(\eta)$ and $y = \rho \sin(\eta)$ for $\rho \in [0, \sqrt{2}]$ and $\eta \in [0, 2\pi]$. This allows us to rewrite $I(r)$ in the following manner

$$I(r) = \int_0^{2\pi} e^{-\frac{(x - r\cos(\alpha))^2 + (y - r\sin(\alpha))^2}{b^2}} d\alpha \tag{28}$$

$$= \int_0^{2\pi} e^{-\frac{(r - \rho\cos(\alpha - \eta))^2}{b^2}} \cdot e^{-\frac{\rho^2 \sin^2(\alpha - \eta)}{b^2}} d\alpha \tag{29}$$

$$= \int_0^{2\pi} e^{-\frac{(r - \rho\cos(\alpha))^2}{b^2}} \cdot e^{-\frac{\rho \sin^2(\alpha)}{b^2}} d\alpha. \tag{30}$$

Setting $t(r) = e^{-\frac{(r - \rho\cos(\alpha))^2}{b^2}}$, we note that this is simply an affine transformation of the Gaussian function $r \mapsto e^{-r^2}$. Consequently, its derivatives can be expressed using Hermite polynomials. Specifically, if $H_k$ is the $k$th Hermite polynomial, we have that

$$t^{(k)}(r) = \frac{(-1)^k}{b^k} H_k \left( \frac{r - \rho\cos(\alpha)}{b} \right) e^{-\frac{(r - \rho\cos(\alpha))^2}{b^2}}. \tag{31}$$

From standard bounds on $H_k$ (DLMF, (18.14.9)), we obtain that

$$|t^{(k)}(r)| \leq \frac{\sqrt{2^k k!}}{b^k}. \tag{32}$$

Computing the $k$th derivative of $I(r)$, plugging in the above bound, and noting that the second factor in the integrand (which does not depend on $r$) is less than one, we obtain

$$|I^{(k)}(r)| \leq \frac{\sqrt{2^k k!}}{b^k}. \tag{33}$$

The Gaussi–Jacobi quadrature error can therefore be bounded by

$$\frac{M!}{2M!} \left( \frac{(M+1)!}{(2M+1)!} \right)^2 \frac{\sqrt{2^{2M} 2M!}}{b^{2M}} = \frac{M!}{\sqrt{2M!}} \left( \frac{(M+1)!}{(2M+1)!} \right)^2 \left( \frac{2}{b^2} \right)^M \tag{34}$$

$$\leq \left( \frac{M!}{2M!} \right)^2 \left( \frac{2}{b^2} \right)^M. \tag{35}$$

Combining these results, we obtain that

$$\sum_{n=0}^{N-1} \sum_{m=0}^{M-1} \frac{w_n}{2\pi M} e^{-\frac{(x - r_n \cos(\alpha_m))^2 + (y - r_n \sin(\alpha_m))^2}{b^2}} \tag{36}$$

$$= \int_D e^{-\frac{(x-u)^2 + (y-v)^2}{b^2}} du dv + O\left( \frac{(\sqrt{2} + \Delta)^2}{N^2 b^4} + \left( \frac{M!}{2M!} \right)^2 \left( \frac{2}{b^2} \right)^M \right) \tag{37}$$

for $D = \{(u, v) \mid u^2 + v^2 < (\sqrt{2} + \Delta)^2\}$. We now need to approximate this integral. To do this, we extend it to an integral over all of $\mathbb{R}^2$, which gives the result $\pi b^2$. To quantify the error in the approximation, we must therefore bound

$$\int_{\mathbb{R}^2 \setminus D} e^{-\frac{(x-u)^2 + (y-v)^2}{b^2}} du dv. \tag{38}$$

We again consider polar coordinates for both $(x, y)$ and $(u, v)$, which transforms the above expression into

$$\int_{\sqrt{2}+\Delta}^{+\infty} \int_0^{2\pi} e^{-\frac{r^2+\rho^2-2r\rho\cos(\alpha-\eta)}{b^2}} \, d\alpha dr = \int_{\sqrt{2}+\Delta}^{+\infty} e^{-\frac{r^2+\rho^2}{b^2}} \int_0^{2\pi} e^{\frac{2r\rho\cos(\alpha)}{b^2}} \, d\alpha dr. \quad (39)$$

The innermost integral can be written as $2\pi I_0(2r\rho/b^2)$, where $I_0$ is the zeroth-order modified Bessel function of the first kind. This gives

$$2\pi \int_{\sqrt{2}+\Delta}^{+\infty} e^{-\frac{r^2+\rho^2}{b^2}} I_0(2r\rho/b^2) dr. \quad (40)$$

Bounding $I_0(2r\rho/b^2)$ by $e^{2r\rho/b^2}$ (DLMF, (10.14.3)), we obtain the upper bound

$$2\pi \int_{\sqrt{2}+\Delta}^{+\infty} e^{-\frac{r^2+\rho^2}{b^2}} e^{\frac{2r\rho}{b^2}} dr = 2\pi \int_{\sqrt{2}+\Delta}^{+\infty} e^{-\frac{(r-\rho)^2}{b^2}} dr = 2\pi \int_{\sqrt{2}+\Delta-\rho}^{+\infty} e^{-\frac{r^2}{b^2}} dr. \quad (41)$$

Since $\rho \leq \sqrt{2}$, we have that $\sqrt{2} + \Delta - \rho \geq 0$, so we can bound this integral using standard results on the integral of a Gaussian function DLMF, (7.8.3) to obtain the bound

$$2\pi b e^{\frac{(\sqrt{2}+\Delta-\rho)^2}{b^2}} \leq 2\pi b e^{-\frac{\Delta^2}{b^2}}. \quad (42)$$

This, together with the quadrature error bounds, gives us that

$$\sum_{n=0}^{N-1} \sum_{m=0}^{M-1} \frac{w_n}{2\pi M} e^{-\frac{(x-r_n\cos(\alpha_m))^2+(y-r_n\sin(\alpha_m))^2}{b^2}} \quad (43)$$

$$= \pi b^2 + O\left( \frac{(\sqrt{2}+\Delta)^2}{N^2 b^4} + \left(\frac{M!}{2M!}\right)^2 \left(\frac{2}{b^2}\right)^M + b e^{-\frac{\Delta^2}{b^2}} \right) \quad (44)$$

Plugging this into our expressions for $M_{pqij}$ and $P^{\mathrm{T}} P x$ then gives us the desired result.

## B    PROPERTIES OF $\phi$

To prove that $\phi[i, j] = (\pi b^2 L^2)^{-1} e^{-\frac{i^2+j^2}{b^2 L^2}}$ sums approximately to one, we use the Poisson summation formula (Stein & Weiss, 1971, Chapter VII.2). First, we note that the Fourier transform of the continuous function

$$\phi(u, v) = \frac{1}{\pi b^2 L^2} e^{-\frac{u^2+v^2}{b^2 L^2}} \quad (45)$$

is given by

$$\widehat{\phi}(\omega, \xi) = e^{-\pi^2 b^2 L^2 (\omega^2+\xi^2)}. \quad (46)$$

We thus have

$$\sum_{i,j=-\infty}^{+\infty} \phi[i, j] = \sum_{i,j=-\infty}^{+\infty} \phi(i, j) \quad (47)$$

$$= \sum_{k,\ell=-\infty}^{+\infty} \widehat{\phi}(k, \ell) \quad (48)$$

$$= 1 + \sum_{k,\ell \neq 0} e^{-\pi^2 b^2 L^2 (k^2+\ell^2)}. \quad (49)$$

Provided that $b$ is large enough, the infinite sum is negligible (for $b = 1/L$, the largest term is of the order $10^{-4}$). We thus have the desired result.

## C   POISSON NOISE PARAMETRIZATION

The (discrete) Poisson distribution has one parameter $\lambda$ and a probability mass function of

$$\text{PMF}_{\text{Pois}(\lambda)}(k) = \lambda^k \frac{e^{-\lambda}}{k!} \tag{50}$$

The mean of this distribution is again $\lambda$, and the standard deviation $\sqrt{\lambda}$.

To use the Poisson distribution directly as a source of shot noise would require comparing concrete finite electron counts, which is inconvenient for comparison with the Gaussian setting as that corresponds to infinite electron rate. Also, the Poisson expectation is always positive, whereas we use by convention noise that is symmetric about zero in the regions where the electron beam is unobstructed by molecules (the background).

We therefore define a slightly different process to emulate shot noise, which has initially two parameters $\eta$ and $\lambda_0$. Given an input signal $x$ which is close to zero in the background, let

$$y = \eta \cdot (\lambda_0 - c), \tag{51}$$

with

$$c \leftarrow \text{Pois}(\lambda_0 - x/\eta). \tag{52}$$

The subtraction $\lambda_0 - x/\eta$ expresses that positive $x$ attenuates the pixel-wise dose, corresponding to blocking of the electrons by the molecule (more accurately, destructive interference due to phase shift). Note that the opposite behavior can be achieved by choosing negative $\eta$.

Then the noisy image has in the background again mean 0, matching the mean in the Gaussian case, and standard deviation

$$N = \sqrt{\lambda_0} \cdot \eta. \tag{53}$$

The local perturbation of the mean due to $x$ has unity gain (because the multiplication by $\eta$ in equation 51 is canceled by division in equation 52).

We want $N$ to match the noise's standard deviation in the Gaussian setting, which is by convention represented as

$$N = \frac{S}{\sqrt{\text{SNR}}}. \tag{54}$$

This can be fulfilled by solving equation 53 for $\lambda_0$, which gives

$$\lambda_0 = \frac{S^2}{\text{SNR} \cdot \eta^2}. \tag{55}$$

$\eta$ remains as a parameter. If $\eta \approx 0$ is chosen, $\lambda_0$ approaches infinity. A Poisson distribution with large $\lambda$ is well approximated by a Gaussian distribution, in that sense the results can directly be compared.

## D   ATTENTION FOR CLUSTERING

The polar transformer architecture is designed to combine information from precisely those image in an input set which match up to plane rotation. It is the process of finding out which ones these are that we call clustering.

How it occurs can be observed by studying the activations in the activation coefficient matrix as a set with known split of two directions is processed. Without loss of generality (because the attention mechanism is equivariant under permutations of the images), this can be realized by taking the first 8 projections in one directions and the remaining 8 projections in another direction. Ideally, the attention (summed over the angular direction) should then have block-matrix structure, as only the images corresponding to the same direction interact with each other. Indeed this matches the observations in the case of relatively high SNR (see Figure 4). Consequently, the model can then also combine information for denoising purposes just as well as if the directions had already been known a priori, as is the case of a directional set. This manifests in MSE scores that are almost as good with two directions as with a single direction (Figure 3).

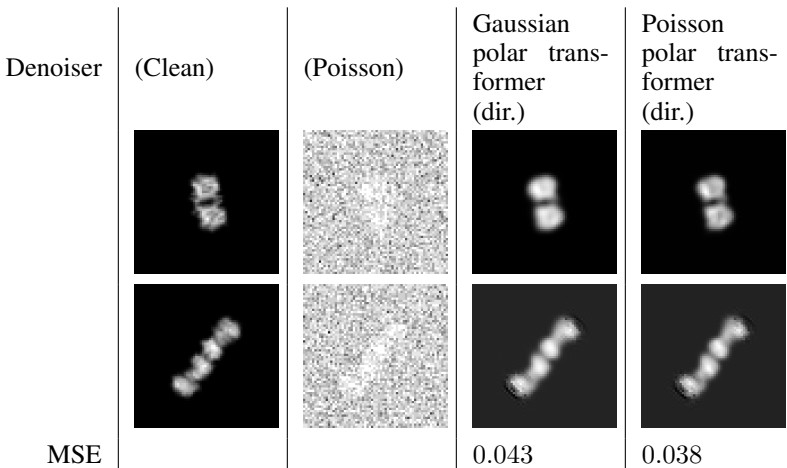

| Denoiser | (Clean) | (Poisson) | Gaussian polar transformer (dir.) | Poisson polar transformer (dir.) |
|---|---|---|---|---|
| MSE | | | 0.043 | 0.038 |

Table 2: Example results like in 1, but for Poisson noise. SNR of $1/32$ by the notion of Equation 54. A transformer model that has been trained on Gaussian noise is compared with one trained on Poisson noise.

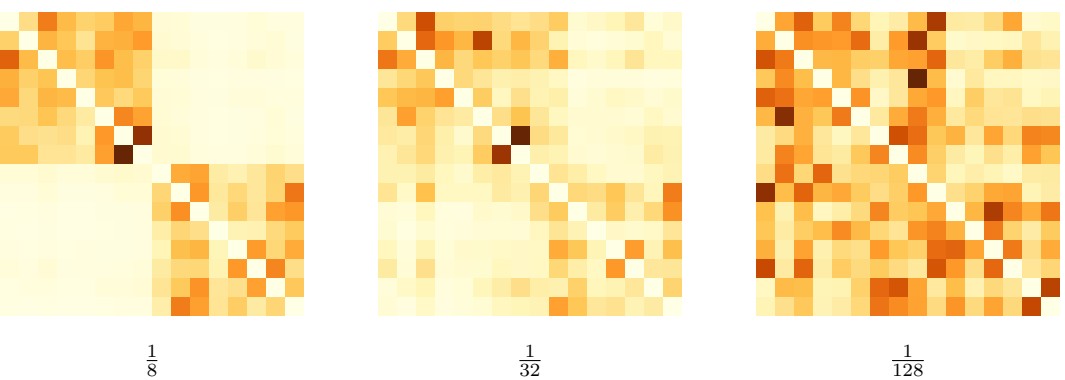

$$\frac{1}{8} \qquad\qquad \frac{1}{32} \qquad\qquad \frac{1}{128}$$

Figure 4: Example attention matrices for a 2-direction, 16-image-set polar transformer model at different SNR levels.

At lower SNR meanwhile, the attention mechanism cannot be as confident in classifying the directions anymore and the block structure of the matrix is more fuzzy. This manifests in MSE values that are not quite as good anymore as in the ideal predetermined case, but there is still more information to be exploited even from the uncertain classification as from only single images and thus even the clustering model still denoises better at low SNR than single-image models can.

## E    EXTENSION TO EXPERIMENTAL DATA

Although this paper works with synthetic data throughout, this is for practical reasons – separation of concerns, comparability with ground truth, possibility to systematically study different SNRs and noise distribution regimes.

By leveraging more accurate simulation models for projection images (Parkhurst et al., 2021) combined with training on projection images from experimental datasets, the proposed architecture can be adapted to realistic datasets. For the latter, the noise2noise paradigm (Lehtinen et al., 2018) has established that it is possible to train denoising models without access to noiseless ground truth, all that is needed are statistically independent noise realisations. Furthermore, cryo-EM data is available in form of videos, from which the required independent noise can be extracted by using alternating video frames (Bepler et al., 2020).

