# OpenReview forum: "Joint Denoising of Cryo-EM Projection Images using Polar Transformers"
_ICLR.cc/2025/Conference — Submitted to ICLR 2025_

### Official Review · Reviewer_34Gd · 2024-10-31

**Soundness:** 2
**Presentation:** 3
**Contribution:** 3
**Rating:** 5
**Confidence:** 3

**Summary:**

This paper deals with the denoising of images produced in the context of cryogenic electron microscopy (cryo-EM) single-particle analysis. In cryo-EM single-particle analysis, an electron microscope records 2D projections of the unknown 3D structure of a single type of particle (e.g., a specific protein) at random orientations. These projections can then be used to estimate a 3D model of the particle. Due to data acquisition limitations, the 2D particle projections have very low signal-to-noise ratios, requiring denoising.

The authors propose two novel deep learning architectures to improve the denoising of 2D particle projections: The Polar CNN and the polar transformer.
- The polar CNN is designed to be equivariant to 2D rotations. The authors argue that this property is desirable because the 2D particle projections capture the particle at random orientations.

- The polar transformer is based on the polar CNN and additionally exploits redundancy in the data by combining information from projection images that show the particle from a similar direction.

The authors test their new architectures on synthetic data and compare them to denoising with a Wiener filter.

**Recommendation:** I do not recommend the paper for publication at ICLR. In my opinion, the weaknesses in the experiments outweigh the reasons to accept the paper (see "Weaknesses" for more details). However, I would like to emphasize that the paper has potential and I am willing to change my evaluation if all my concerns regarding the experiments are sufficiently addressed.

**Post rebuttal:** I have increased my score from **3 to 5**.  As part of their rebuttal, the authors have clarified the details of the angular attention mechanism. Moreover, they have included a comparison to a deep learning-based denoiser in which they show that the polar transformer architecture outperforms "vanilla" networks. Overall, I think that the proposed architectures are interesting contributions.
I have not increased my score further because one of the main concerns I described in my review remains: The authors have not tested their methods on real data which comes with additional challenges such as the need for self-supervised training due to the lack of ground truth.

**Strengths:**

- The use of architectures that are rotationally equivariant and explicitly exploit redundancy for denoising individual particle projections is a novel and interesting idea and has the potential to improve upon existing deep learning-based denoising methods.
- Experiments on synthetic data suggest that exploiting the redundancy in the data (as done by the polar transformer in the "directional" setup) is beneficial to the denoising performance.
- The paper is clearly written in most parts (only section 5.2 is a bit unclear, see "Weaknesses").

**Weaknesses:**

Major Weaknesses:


- Weaknesses in the experiments (Section 6):

	- The authors only compare their polar CNN and their polar transformer to a denoising Wiener filter, and there is no comparison to other deep learning-based denoisers, which typically outperform classical denoising methods. Good candidates for deep learning based baselines could be, e.g., CryoCARE [1] and Topaz [2].

	- There is no experiment that compares the denoising performance of a standard 2D convolutional network with that of a polar CNN and a polar transformer.
		Therefore, it is unclear how much denoising actually benefits from the equivariance properties of the polar CNN and the polar transformer.

	- The authors train their polar CNN and their polar transformer in a supervised manner. In my opinion, this setup is not realistic, since pairs of noisy images and clean underlying ground truth are typically not readily available in cryo-EM single-particle analysis (see also "Questions").
		Perhaps it would be possible for the authors to consider self-supervised training without clean ground truth for their experiments. (This has already been done for methods such as CryoCARE and Topaz).

	- All experiments are on synthetic data. There are no experiments on real images produced with cryo-EM. This recent dataset proposed by Dhakal et al. [3] contains real cryo-EM images with picked particles. Maybe this data can be used for further evaluation.


- Section 5.2 on angular attention, which is one of the central contributions of the paper, is unclear to me. There is no explicit definition of the angular attention layer $f_\theta^{\text{ang-attention}}$. I have formulated some questions about angular attention in the "Questions" section.

Minor weaknesses:
- It is unclear to me which parts of the polar CNN and polar transformer architectures are novel and which are from existing work. Perhaps the authors could make this clearer in the paper.
- This is more of a suggestion: Figures illustrating angular convolution and angular attention mechanisms might make these concepts a bit easier for readers to grasp.




--------------------------------------------
[1] Buchholz, Tim-Oliver, et al. "Cryo-CARE: Content-aware image restoration for cryo-transmission electron microscopy data." 2019 IEEE 16th International Symposium on Biomedical Imaging (ISBI 2019). IEEE, 2019.

[2] Bepler, Tristan, et al. "Topaz-Denoise: general deep denoising models for cryoEM and cryoET." Nature communications 11.1 (2020): 5208.

[3] Dhakal, Ashwin, et al. "A large expert-curated cryo-EM image dataset for machine learning protein particle picking." Scientific Data 10.1 (2023): 392.

**Questions:**

- Given the challenges associated with supervised training of a denoiser (see "Weaknesses"), how would the authors train a model that can be used to denoise real, experimental images? Is it possible for the model to generalize well from synthetic to real data?

- In lines 63 - 65, the authors state that denoising a micrograph is easier than denoising selected projections of individual particles. If this is true, wouldn't a method that denoises the entire micrograph *before* picking perform better than the authors' proposed approach of denoising individual particle projections?

- Questions about angular attention:
	- Why are only the keys rotated by the angle $\alpha_\ell$? (line 373-374)
	- The authors state that "[they] apply a corresponding rotation to the value vectors". Does this require knowing the angle $\alpha_\ell$? If so, why can the angle be assumed to be known?
	- Could the authors please provide an explicit formula or algorithm for the angular attention mechanism?

- Why is the value net in the polar transformer "set to identity" (line 417)?

---

> ### Author Response · Authors · 2024-11-27
>
> Weaknesses
> ===
>
> > The authors only compare their polar CNN and their polar transformer to a denoising Wiener filter, and there is no comparison to other deep learning-based denoisers...
>
> Existing cryo-EM denoisers are target a somewhat different problem, that of denoising of entire micrographs.
> The use case is also somewhat different, in that these mainly serve to allow for visual inspection or to improve particle picking.
> Furthermore, given the different modalities, any comparison would need to be done in an integrated setting, comparing not only the denoising processes themselves but pipelines that can apply them to the same data.
>
> > There is no experiment that compares the denoising performance of a standard 2D convolutional network with that of a polar CNN and a polar transformer...
>
> See overall response.
>
> > The authors train their polar CNN and their polar transformer in a supervised manner. In my opinion, this setup is not realistic, since pairs of noisy images and clean underlying ground truth are typically not readily available...
>
> Please see response to questions below.
>
> > All experiments are on synthetic data. There are no experiments on real images produced with cryo-EM...
>
> We have looked into the suggested dataset. Ultimately our model needs to prove itself on such inputs, but this first requires training on experimental data (as outlined in the main response, see above), which is currently not done.
> Another issue is that the current implementation of the network only works for smaller set of images and cannot therefore handle a full cryo-EM dataset.
>
> Overall, we agree that training and testing on experimental data is necessary.
> That being said, this manuscript provides a proof of concept of the central mechanisms and validates those on synthetic data, providing a basis for future work in adapting it to an experimental setting.
> Given the novelty of the architecture and the improvement over standard baselines, we believe this constitutes a relevant contribution to the literature on its own.
>
> > Section 5.2 on angular attention ... is unclear to me.
>
> Please see response to questions below.
>
> > It is unclear to me which parts of the polar CNN and polar transformer architectures are novel...
>
> We have not found a RBF-based polar representation like the one proposed in this work elsewhere in the literature.
> The polar transformer has not been presented elsewhere to the best of our knowledge.
> We have highlighted this better in the introduction.
>
> > This is more of a suggestion: Figures illustrating angular convolution and angular attention mechanisms...
>
> We agree that this would be useful, but due to time constraints, we have not been able to incorporate this.
>
> Questions:
> ===
>
> > How would the authors train a model that can be used to denoise real, experimental images?
>
> We plan to train our models (either complete and parts of synthetically-pretrained ones) in a noise2noise fashion.
> We are confident that the proposed architecture has no fundamental roadblocks for this kind of training, but there are technical challenges in the implementation of the required pipeline.
>
> We added an appendix section that provides some discussion on the matter.
>
> > The authors state that denoising a micrograph is easier than denoising selected projections of individual particles.
>
> This sentence was sloppily formulated and we have updated it.
> The point is that a micrograph contains redundancy -- the redundancy that our work uses to obtain better results when denoising sets rather than single projections.
> Micrograph denoising is thus simpler compared to single-projection denoising since it can use information from multiple projections (in the same micrograph) to denoise.
> However, it is also limited to denoising a single micrograph at a time, which means it cannot take advantage of information from multiple micrographs.
>
> Questions about angular attention:
> ---
>
> > Why are only the keys rotated?
>
> The purpose of the rotation is to match the different images in the set, rotation-wise. Rotating both the keys and the queries would not accomplish that, it is about the differing _relative_ rotation.
>
> > The authors state that "[they] apply a corresponding rotation to the value vectors". Does this require knowing the angle?
>
> It does not require the angle to be known. The method averages over all possible rotations, weighted by the angular attention coefficients (which are in turn generated by the convolution of key and query vectors).
>
> > Could the authors please provide an explicit formula or algorithm for the angular attention mechanism?
>
> This was an oversight in the initial submission, we have now made the definition explicit in equations 14 and 15.
>
> > Why is the value net in the polar transformer "set to identity"?
>
> An intuitive reason is that the attention mechanism mimics class averaging, which combines the original images by rotating and averaging.
> That being said, a learned value net can be used here, but performance remains the same.

---

> > ### Comment · Reviewer_34Gd · 2024-11-29
> >
> > I thank the authors for their detailed answers and their clarifications. I have increased my score and updated my review.

---

### Official Review · Reviewer_UvAR · 2024-11-03

**Soundness:** 3
**Presentation:** 4
**Contribution:** 3
**Rating:** 6
**Confidence:** 3

**Summary:**

This paper addresses denoising in cryo-EM images by designing a neural network that is equivariant to rotations. The authors achieve this by constructing a polar representation of the image and applying convolutions in this transformed space, thereby effectively converting the translational equivariance of standard convolutions into rotational equivariance. These _polar CNNs_ then serve as the key and query networks in a transformer to obtain a rotation-equivariant attention mechanism between images that should allow for solution of the full set denoising task, i.e. the implicit clustering of images by viewing angle for an improved denoising performance.

**Strengths:**

The paper is clearly written and provides a great introduction to the problem of denoising in cryo-EM reconstruction. The authors introduce their method step-by-step, making it easy to follow the rationale of each of their design choices.

The use of a polar representation to achieve rotational equivariance is innovative and well-founded, facilitating deep learning-based denoising that can leverage the signal from images taken at the same viewing angle without requiring an explicit preprocessing step to cluster by angle.

**Weaknesses:**

While the theoretical foundation of the method is convincing, the experimental validation of its effectiveness is lacking.

The only baseline comparison is to Wiener filtering, with no comparisons to other deep learning-based methods, despite several being mentioned in the related work section. I suggest to include a comparison to at least one of the deep-learning based methods, such as the DnCNN (Zhang et al., 2017) or a U-Net denoiser adapted for cryoEM data.

Further, all experiments are conducted on simulated data. It therefore remains unclear whether the method would generalize to real data acquisitions as well. Do the authors envision the method to be trained on simulated data and then be applied to real data? In particular, the training and evaluation both use sets of images that only contain two viewing angles. How can this be achieved for real data? This apparent limitation is not discussed. I suggest the authors either 1) include experiments where they apply their method to real data or 2) discuss challenges they foresee in the application to real data, including possible domain shift and a random sampling of viewing angles.

In conclusion, the paper could be strengthened significantly by an extended set of experiments that show this theoretically-compelling method to work well in practice and a discussion of its real-world limitations. Specifically, I suggest the authors include a deep-learning based method as an additional baseline to make their evaluation more comprehensive and demonstrate or discuss the applicability of their method to real data. If application to real data is currently not feasible, I suggest to include a discussion of those limitations and an outline for future work for how those limitations could potentially be overcome.

**Questions:**

Questions:
- Are the weights shared between key and query networks?
- The conclusion mentions that important future work includes scaling the method to larger datasets. Which part of the method does not scale right now?

Typos/Format:
- L142: providing a stronger priors -> providing stronger priors
- L229/230: -L/2, ..., -L/2-1 -> -L/2, ... , L/2 -1
- L269: fourier transform of $\phi$?
- L464: reference not resolved

---

> ### Author Response · Authors · 2024-11-27
>
> Weaknesses
> ===
>
> > While the theoretical foundation of the method is convincing, the experimental validation of its effectiveness is lacking.
> > The only baseline comparison is to Wiener filtering, with no comparisons to other deep learning-based methods, despite several being mentioned in the related work section.
> > I suggest to include a comparison to at least one of the deep-learning based methods, such as the DnCNN (Zhang et al., 2017) or a U-Net denoiser adapted for cryoEM data.
>
> The main reason we did not compare to the cited deep learning methods (such as Topaz-Denoise) is that they solve a different problem: denoising entire micrographs as large single images, whereas we denoise sets of already picked molecules (see overall response for more details).
> For the DnCNN and U-Net methods, we have added results in the manuscript, showing that the proposed polar transformer outperforms them by a significant margin (being able to incorporate information from multiple images instead of being limited to a single projection image).
>
> > Further, all experiments are conducted on simulated data. It therefore remains unclear whether the method would generalize to real data acquisitions as well. Do the authors envision the method to be trained on simulated data and then be applied to real data? In particular, the training and evaluation both use sets of images that only contain two viewing angles. How can this be achieved for real data? This apparent limitation is not discussed. I suggest the authors either 1) include experiments where they apply their method to real data or 2) discuss challenges they foresee in the application to real data, including possible domain shift and a random sampling of viewing angles.
>
> Training on simulated data but applying to real is certainly one option, but we also plan to venture into self-supervised training in the form of noise2noise.
> There is no actual restriction to two viewing directions (which would indeed be a severe limitation for practical use).
> The reason we have that in the show results is that the current implementation is very intensive on GPU memory, which limits the size of sets that can be passed into the model.
>
> We have expanded information about how the method can be adapted for applications to real data in the conclusion and an additional appendix.
>
> Questions
> ===
>
> > Are the weights shared between key and query networks?
>
> Yes the weights are the same. We have not observed a large difference by allowing key and query vectors to have different work. This has been clarified in the manuscript.
>
> > The conclusion mentions that important future work includes scaling the method to larger datasets. Which part of the method does not scale right now?
>
> See the above point about set size and number of viewing directions.
>
> Typos/Format
> ===
>
> Fixed.

---

> > ### Comment · Reviewer_UvAR · 2024-12-02
> >
> > I thank the authors for their rebuttal.
> >
> > The major concerns raised in my and other reviews have been addressed and I have updated my score accordingly.
> >
> > I still think that it remains unclear whether the method will perform well on real data, particularly for the full denoising task, where the current experiments do not fully reflect those conditions (i.e., will the transformer generalize to an arbitrary distribution of viewing angles?). Those challenges could be discussed further in the added appendix section.
> > However, I acknowledge that such experiments are outside the scope of this manuscript and do not think it should preclude it from being accepted. I hope the authors will be able to demonstrate the applicability to real data in future work.

---

### Official Review · Reviewer_cDLY · 2024-11-04

**Soundness:** 2
**Presentation:** 2
**Contribution:** 2
**Rating:** 6
**Confidence:** 3

**Summary:**

This paper presents a deep learning approach to denoise cryo-EM images, which is challenging due to high noise levels and varied viewing directions. The authors apply polar representation within both proposed CNN and transformer models, achieving improved results in denoising.

**Strengths:**

1. Leveraging polar representation can simplify the denoising task by translating rotational variance into angular shifts. This approach is promising for denoising given the rotational nature of cryo-EM.
2. Using a dataset of 5,000,000 images strengthens the model's robustness and enables it to learn effectively from diverse viewing angles.
3. Testing both Gaussian and Poisson noise further validates the proposed model.
4. Testing both CNN and transformer architectures increases the reliability of the proposed method.

**Weaknesses:**

1. Some key details in the approximations lack mathematical measurements (see Q1).
2. Results are only compared to the Wiener filter method. Including comparisons with more state-of-the-art methods would be better.
3. Adding a mathematical proof explaining why and how polar representation improves denoising accuracy would strengthen the theoretical foundation of the approach (Related to Q3).
4. At leat two typos. (See Q5).

**Questions:**

1. Could you provide more details on the approximations in Eq. (8) and Eq. (10), such as the error order?

2. Will your model work with negative binomial noise?

3. You stated that "f(R_alpha(x)) = R_alpha(f(x))" when f could be a statistical noise distribution. If we consider the denoising model H_theta as an inverse of a statistical noise distribution, would "H_theta(R_alpha(x)) = R_alpha(H_theta(x))"? If so, how does the polar representation improve accuracy? If not, could you explain the key difference?

4. In Section 5.2, I think "f_key" and "f_query" are symmetric. That is, using "m - l" in either should work. So, why is f_key chosen over f_query? I also wonder if adding both query and key with the same value, but with output as "S_l = (sqrt(2) / 2 * S_l_key) * (sqrt(2) / 2 * S_l_query)" would yield more accurate results, since it allows the model more freedom to learn more features.

5. On line 178 and 179, should “the” be changed to “they”? On line 229 and 230, should it be "L/2 - 1"? And on line 464 and 465, there is an a question mark.

---

> ### Author Response · Authors · 2024-11-27
>
> Weaknesses
> ===
>
> > 1. Some key details in the approximations lack mathematical measurements (see Q1).
>
> We have now strengthened the central mathematical result regarding the invertibility of the polar representation by providing rigorous error bounds on the approximation of the $P^T P$ operator by a convolution. This is backed up by a proof in the appendix. We see that for large enough values of $N$ and $M$ (the number of polar grid points), we obtain a very accurate approximation, a fact which is borne out by the low reconstruction error shown in Figure 1.
>
> > 2. Results are only compared to the Wiener filter method. Including comparisons with more state-of-the-art methods would be better.
>
> We added comparisons to a DnCNN and a U-Net. The caveat here is that such methods are in practice used in a way different from ours: e.g. Topaz would apply a denoiser to an entire micrograph containing many copies of the molecule.
> The paradigm of our method meanwhile employs multiple images in the form of sets, which the standard methods have no way to do (see overall response).
>
> > 3. Adding a mathematical proof explaining why and how polar representation improves denoising accuracy would strengthen the theoretical foundation of the approach (Related to Q3).
>
> The polar representation per se does not much improve denoising accuracy (only a little, because rotational equivariance is baked-in then whereas cartesian methods would have to learn it from training data).
> Polar coordinates are however a requirement for our method because it allows computing the angular attention mechanism in terms of an FFT. Without that, this mechanism would be unusably slow in practice.
>
> > 4. At leat two typos. (See Q5).
>
> Thank you. These have been fixed.
>
> Questions
> ===
>
> > 1. Could you provide more details on the approximations in Eq. (8) and Eq. (10), such as the error order?
>
> As stated above, we have now given error bounds on eq. (8) along with a proof of these bounds. Regarding eq. (10), this is not possible to do without a rigorous definition of the rotation operator applied to images. This can be done in several ways (nearest-neighbor interpolation, linear interpolation, etc.), some of which will be more compatible with the polar representation compared to others. Picking a definition of rotation and determining its relationship to the polar representation quickly becomes quite technical and we have therefore considered it to be outside the scope of our current work.
>
> > 2. Will your model work with negative binomial noise?
>
> We have not tried this, but this would correspond to an overdispersed variant of the the Poisson noise setup (see Appendix C) and the model does not appear to latch on the discrete counting levels.
> We are thus fairly confident that it would also work with negative binomial noise.
>
> > 3. You stated that "f(R_alpha(x)) = R_alpha(f(x))" when f could be a statistical noise distribution. If we consider the denoising model H_theta as an inverse of a statistical noise distribution, would "H_theta(R_alpha(x)) = R_alpha(H_theta(x))"? If so, how does the polar representation improve accuracy? If not, could you explain the key difference?
>
> It is not clear to us what is meant here. The function $f$ is the denoiser. The statistical distribution is invariant under rotations (i.e., if $\Pi(x)$ is the distribution of images, we have $\Pi(R_\alpha(x)) = \Pi(x)$). The denoiser needs to be equivariant in order for the distribution of denoised images also to be invariant under rotations.
>
> > 4. In Section 5.2, I think "f_key" and "f_query" are symmetric. That is, using "m - l" in either should work. So, why is f_key chosen over f_query?
>
> This is an arbitrary convention. The method could also be written in other ways, but the crucial part is that there is a relative rotation between keys and queries.
>
> > I also wonder if adding both query and key with the same value, but with output as "S_l = (sqrt(2) / 2 * S_l_key) * (sqrt(2) / 2 * S_l_query)" would yield more accurate results, since it allows the model more freedom to learn more features.
>
> We are unsure about what is meant here.
>
> > On line 178 and 179, should “the” be changed to “they”? On line 229 and 230, should it be "L/2 - 1"? And on line 464 and 465, there is an a question mark.
>
> Thank you. These have been fixed.

---

### Official Review · Reviewer_BAWp · 2024-11-07

**Soundness:** 2
**Presentation:** 2
**Contribution:** 2
**Rating:** 3
**Confidence:** 4

**Summary:**

This paper develops a strategy for jointly denoising single particle cyroEM particle projections. The proposed approach is based upon applying a CNN (to one image) or a transformer architecture (to many images) that is/are transformed into polar coordinates. The network architectures are specialized to have desirable invariances in the polar domains. The proposed method is tested on simulated data with varying amount of Poisson and Gaussian noise. The proposed method outperforms simple Weiner filtering.

**Strengths:**

+++Proposed method (using an invariant network to jointly denoise multiple images in the polar domain) is novel and well-motivated

**Weaknesses:**

---Contextualization/Practical relevance: The manuscript states the single particle reconstruction pipeline "involves first extracting images of individual particles from the micrographs, denoising these images, estimating their corresponding viewing directions, and then reconstructing a 3D density map". However, it is my understanding that modern reconstruction techniques (e.g., expectation maximization [A]) marginalize over the various view directions and avoid denoising the 2D projections all together.
The manuscript mentions contamination detection as another possible use case for denoising, which may be valid. The manuscript needs to clarify which parts of the reconstruction pipeline it may and may not improve.

[A] Scheres, Sjors HW. "RELION: implementation of a Bayesian approach to cryo-EM structure determination." Journal of structural biology 180.3 (2012): 519-530.


---Weak baselines: The proposed method outperforms Weiner filtering by a large margin. How does it compare to advanced non-learning-based methods like BM3D? How does it compare to a conventional CNN without a polar transformation?

---Missing ablations: There is no evidence any of the proposed network choices are useful. How does the proposed method compare to a single U-net applied to cartesian coordinates? How important are the various architectural choices?

---Section 1 says the proposed method clusters, aligns, and denoises but the next section states it is assumed the images have been centered. If so, why is alignment necessary?

----How does proposed method compare to denoising entire micrographs?

---Typo: L464 a reference is missing

---Ignores CTF effects. To the paper's credit, this limitation is clearly acknowledged.

**Questions:**

How relevant is the proposed contributions in modern single particle cryoEM pipelines? Do denoisers have a role to play?

How does the proposed method compare to high-performance image denoisers?

Which of the design choices (e.g, polar coordinates) matter and which don't?

---

> ### Author Response · Authors · 2024-11-27
>
> Weaknesses
> ===
>
> > It is my understanding that modern reconstruction techniques (e.g., expectation maximization [A]) marginalize over the various view directions and avoid denoising the 2D projections all together.
>
> While it is correct that EM-based reconstruction methods do not require denoising, several other tasks in cryo-EM image processing do (see below).
> In particular, the set of input images to EM refinement in RELION must be sorted to identify high-quality images.
> Within the RELION reconstruction pipeline, this is typically done in the form of class averaging of the initial picked particles images (section 2.2 in [Scheres, 2015](https://dx.doi.org/10.1016/j.jsb.2014.11.010) and the [RELION tutorial](https://relion.readthedocs.io/en/release-4.0/SPA_tutorial/Class2D.html)).
> Our method can be seen as a different type of class averaging method that has the advantage of incorporating prior information from the training on other known molecules.
>
> > The manuscript mentions contamination detection as another possible use case for denoising, which may be valid. The manuscript needs to clarify which parts of the reconstruction pipeline it may and may not improve.
>
> We have added more examples and references for the use of denoising in the cryo-EM reconstruction pipeline.
>
> > Weak baselines: The proposed method outperforms Weiner filtering by a large margin. How does it compare to advanced non-learning-based methods like BM3D? How does it compare to a conventional CNN without a polar transformation?
>
> We added a comparison with two Cartesian CNN models. Both perform almost as well as the polar CNN, and substantially worse than the transformer models.
>
> BM3D seems unlikely to be competitive with any of the other methods, since it relies on repetitive patterns within each input image, but the raw images are almost exclusively noise and even the denoised version of each of them shows only one molecule projection with very little repeated material.
>
> > How does the proposed method compare to a single U-net applied to Cartesian coordinates? How important are the various architectural choices?
>
> We have added comparison to a U-net and DnCNN. Both perform almost as well as the polar CNN.
> The conclusion is that the polar representation is not very impactful by itself (as equivariance can also be learned), but it is prerequisite for the implementation of the angular attention mechanism and the polar transformer, which is the central contribution of the paper.
>
> > Section 1 says the proposed method clusters, aligns, and denoises but the next section states it is assumed the images have been centered. If so, why is alignment necessary?
>
> Centering here refers to translational centering, while alignment referes to rotational alignment.
> We have made this more explicit in the text.
> (Centering of noisy images can be achieved by other methods, such as [Heimowitz et al., 2021](https://doi.org/10.1137/20M1365946).)
>
> > How does proposed method compare to denoising entire micrographs?
>
> As discussd in the overall response and the response to reviewer 34Gd, the micrograph-denoising problem and the projection-denoising problem are quite distinct with different challenges, making it hard to perform a meaningful comparison between the two.
>
> > Typo: L464 a reference is missing
>
> Thank you. This as been fixed.
>
> > Ignores CTF effects. To the paper's credit, this limitation is clearly acknowledged.
>
> The effect of CTFs will indeed need to be considered in future work. Our expectation is that this will not require dramatic changes to the architecture, since CNNs are quite capable of learning deconvolution. For the purposes of this work, demonstrating the concept of the polar transformer, however, we believe that a testing on data without CTFs is sufficient.
>
> Questions
> ===
>
> > How relevant is the proposed contributions in modern single particle cryoEM pipelines? Do denoisers have a role to play?
>
> We think they do.
> Denoising is beneficial for a range of further processing tasks (see above and the revised introduction) and denoised images are well suited for human inspection.
> Furthermore, a neural denoising network provides an important first component in a learned end-to-end reconstruction model which takes raw projection images as input and generates a 3D reconstruction as output.
>
> > How does the proposed method compare to high-performance image denoisers?
>
> At least within our synthetic data scenario we outperform all plausible contenders.
>
> > Which of the design choices (e.g, polar coordinates) matter and which don't?
>
> The angular attention mechanism is the crucial component to the success of the model.
> The polar representation is a prerequisite for the implementation of this mechanism (besides minor improvements when used by itself, due to baked-in equivariance).

---

### Author Response · Authors · 2024-11-22
**Response to common issues**

First, we would like to thank the reviewers for their thorough reviews.
We are currently in the process of updating the manuscript and providing detailed responses to each reviewer.
In the meantime, however, we wanted to discuss some of the most important issues raised by several reviewers.

The main criticism is that we did not compare with state-of-the-art methods for denoising, such as DnCNNs and U-Nets.
A related issue is the lack of comparison against other methods for cryo-EM denoising in particular, such as Topaz-Denoise and Cryo-CARE.

**Baseline methods**. Regarding the former, we have conducted experiments by training the DnCNN architecture and a standard U-Net on the same training data (single images) and included the results in the revised manuscript.
These indicate a performance similar to the polar CNN, which is not very surprising given the large training set and implicit data augmentation given the large set of viewing directions provided (other experiments using training sets with sparser sets of viewing directions results in improved performance for the polar CNNs due to their built-in equivariance).
We do, however, still see that the polar transformer networks, which denoise multiple images simultaneously, perform significantly better than these single-image denoisers.
Indeed, the central novelty of the paper is not the equivariance of the polar CNNs per se, but its use as a building block for the polar transformer.
We therefore make no strong claim about the performance of the polar CNN with respect to other denoisers.
In the revised manuscript, we have attempted to highlight this crucial distinction better.

**Other cryo-EM denoising methods**. On the comparison with other cryo-EM denoising methods such as Topaz-Denoise and Cryo-CARE, the issue is that these methods attempt to solve a different problem than our proposed method.
In particular, both Topaz-Denoise and Cryo-CARE work by denoising entire micrographs at a time.
For our proposed method, we denoise images of already picked particles that have already been extracted from the micrographs.
This brings with it specific challenges (less information) and opportunities (more structure) compared to the micrograph-denoising task, and thus is not comparable in a meaningful way.

**Experimental data**. Another issue raised by many reviewers is the lack of validation on experimental data.
We agree that this is an important problem and one that we are working on solving.
As suggested by some of the reviewers, self-supervised learning is most likely a reasonable approach here using training protocols such as noise2noise.
More accurate synthetic data, such as those generated by Parakeet (Parkhurst et al., 2021), would also result in more robust models.
However, the scope of this work is to show how the basic architecture of such a joint denoising network can be constructed and its performance verified on experimental data.
There are, of course, many issues to solve before we can present results on experimental data (construction of a curated, high-quality database of experimental projection images being an important one), but we believe that the current manuscript provides a strong proof of concept for this idea.

---

> ### Author Response · Authors · 2024-11-27
>
> We have now uploaded the revised manuscript.

---

### Meta-Review · Area_Chair_6kWB · 2024-12-19

**Metareview:**

The paper proposes a deep learning based method for simultaneous clustering, alignment, and averaging to denoise cryo-EM data.

The paper received mixed reviews. A key issue, pointed out by several reviewers is that while the theoretical contribution is interesting, the proposed method promising, and the theory convincing (reviewer UvAR), the experimental validation is not. Several reviewers pointed out that the baselines considered are not sufficient, and that all experiments are only on simulated data. In the rebuttal, the authors added comparisons to standard denoising baselines, which addressed reviewer UvAR's concern, but not that of other reviewers (Reviewer 34Gd).

I agree with the reviewers that the paper proposes a potentially promising method, however a more thorough experimental evaluation including results on real data are required for acceptance.

**Additional Comments On Reviewer Discussion:**

see above

---

### Decision · Program_Chairs · 2025-01-22

Reject